# Conformational Heterogeneity and Cooperative Effects of Mammalian ALOX15

**DOI:** 10.3390/ijms22063285

**Published:** 2021-03-23

**Authors:** Igor Ivanov, Alejandro Cruz, Alexander Zhuravlev, Almerinda Di Venere, Eleonora Nicolai, Sabine Stehling, José M. Lluch, Àngels González-Lafont, Hartmut Kuhn

**Affiliations:** 1Lomonosov Institute of Fine Chemical Technologies, MIREA—Russian Technological University, Vernadskogo pr. 86, 119571 Moscow, Russia; igor_ivanov@gmx.de (I.I.); alekszhur95@yandex.ru (A.Z.); 2Departament de Química, Universitat Autònoma de Barcelona, 08193 Bellaterra, Barcelona, Spain; Alejandro.Cruz@uab.cat (A.C.); JoseMaria.Lluch@uab.cat (J.M.L.); Angels.Gonzalez@uab.cat (À.G.-L.); 3Department of Experimental Medicine, University of Tor Vergata, Via Montpellier 1, 00133 Rome, Italy; divenere@med.uniroma2.it (A.D.V.); nicolai@med.uniroma2.it (E.N.); 4Institute of Biochemistry, Charite—University Medicine Berlin, Corporate member of Free University Berlin, Humboldt University Berlin and Berlin Institute of Health, Charitéplatz 1, D-10117 Berlin, Germany; sabine.stehling@charite.de; 5Institut de Biotecnologia i de Biomedicina (IBB), Universitat Autònoma de Barcelona, 08193 Bellaterra, Barcelona, Spain

**Keywords:** lipoxygenases, crystal structure, protein–protein interactions, cooperative effects, molecular dynamics

## Abstract

Arachidonic acid lipoxygenases (ALOXs) have been suggested to function as monomeric enzymes, but more recent data on rabbit ALOX15 indicated that there is a dynamic monomer-dimer equilibrium in aqueous solution. In the presence of an active site ligand (the ALOX15 inhibitor RS7) rabbit ALOX15 was crystalized as heterodimer and the X-ray coordinates of the two monomers within the dimer exhibit subtle structural differences. Using native polyacrylamide electrophoresis, we here observed that highly purified and predominantly monomeric rabbit ALOX15 and human ALOX15B are present in two conformers with distinct electrophoretic mobilities. In silico docking studies, molecular dynamics simulations, site directed mutagenesis experiments and kinetic measurements suggested that in aqueous solutions the two enzymes exhibit motional flexibility, which may impact the enzymatic properties.

## 1. Introduction

Arachidonic acid lipoxygenases (ALOXs) isoforms form a family of lipid peroxidising enzymes, which metabolize free and esterified polyunsaturated fatty acids to bioactive hormones commonly named eicosanoids [1,2,3]. In the human genome 6 functional ALOX genes (*ALOX15, ALOX15B, ALOX12, ALOX12B, ALOXE3, ALOX5*), several dysfunctional pseudogenes and a single *ALOX12* antisense gene (*ALOX12-AS1*) with unexplored biological relevance have been identified [4]. The corresponding enzymes have been implicated in cell development and maturation, but also play a patho-physiological role in different inflammatory, hyperproliferative and neurological disorders [1,5]. Traditionally, ALOX isoforms have been considered as monomeric enzymes containing a single substrate binding pocket, in which fatty acids are bound in close proximity to the catalytically active non-heme iron. Detailed kinetic experiments on the oxygenation of different fatty acid substrates indicated that the presence of lipoxygenase products such as 15-H(p)ETE, 12-H(p)ETE, and 13-H(p)ODE modulate the kinetic constants of arachidonic acid (AA) and linoleic acid (LA) oxygenation [6,7,8] by ALOX15. Moreover, the potency of several ALOX15 inhibitors depend on the chemistry of the substrate fatty acid employed for the activity assays [9,10]. These data are not consistent with the conventional mechanism of the ALOX reaction and strongly suggest allosteric mechanisms. Unfortunately, the molecular basis for the allosteric properties of ALOX-isoforms remains unclear but two alternative concepts have been suggested: (i) Allosteric ALOX-isoforms involve, in addition to the substrate-binding pocket, a binding site for allosteric regulators. Ligand binding at this hypothetic motif alters the structure of the substrate-binding pocket and thus, the kinetics of substrate oxygenation [11,12]. (ii) Allosteric ALOX isoforms may function as dimeric proteins and one monomer within these dimers acts as allosteric regulator [13]. In this scenario, the substrate binding pocket of the regulatory monomer constitutes the binding site for the allosteric ligand, whereas the substrate fatty acid is bound in the pocket of catalytic monomer.

In crystals rabbit ALOX15 (PDB entry 2P0M) is present as a mixed protein dimer, in which two structurally different conformers A and B interact with each other via their α2 and α18 helices [14]. Monomer B involves an active site ligand (RS7 inhibitor) inside the putative substrate binding pocket, whereas the corresponding structural motif of monomer A is empty. Except for the α2 and α18 helices the spatial arrangement of the other secondary structural elements of the two conformers A and B is very similar. These two helices directly interact with each other stabilizing the heterodimer. In the liganded conformer B α18 helix as a whole is slightly dislocated when compared with its counterpart in the ligand-free conformer A. On the other hand, α2 helix in the liganded conformer B is strongly displaced by about 12 Å when compared with the non-liganded monomer A [14]. Free-energy calculations on rabbit ALOX15 dimerization suggested that among the possible ALOX15 dimers (A+A, A+B, B+B) the A+B combination is thermodynamically most stable [15].

In general, in aqueous solution the motional flexibility of the secondary structural elements is less limited when compared with crystals [15]. Under these conditions rabbit ALOX15 also undergoes reversible dimerization, in which non covalently linked dimers are formed. In contrast, human ALOX5 forms covalently linked dimers, in which monomers are interconnected via disulfide bridges [16]. The degree of rabbit ALOX15 dimerization depends on protein concentration, temperature, pH, ionic strength and presence or absence of active site ligands [13,15]. In low ionic strength buffers (20 mM Tris pH 6.8 to 8.0) a linear correlation between the monomer-dimer ratio and the enzyme concentration was observed [15], but under hypertonic conditions (200 mM NaCl) a higher degree of protein dimerization was detected. This shift was associated with a high level of flexibility of the regulatory PLAT domain [15]. Amino acid residues located at the end of the α18 helix (I593, L597) contribute to the substrate-binding cavity and may be relevant for proper substrate alignment [17,18,19]. In fact, previous efforts to dock AA or LA into the substrate binding pocket of conformer A resulted in unstable enzyme-substrate complexes (unpublished data) and these data suggest that rearrangement of the α18 helix might be required to provide sufficient space for fatty acid binding. However, the driving forces for the rearrangement of α2 helix remain unclear. Since a monomer-dimer equilibrium has also been observed for the ligand free enzyme, it might be possible that rabbit ALOX15 in aqueous solutions may also be present in two conformational states, which differ from each other by the localization of their α2 helices. Whether the displacement of α2 helix is forced by RS7 binding or whether the intrinsic motional flexibility of α2 helix allows RS7 binding cannot be decided on the basis of the currently available data. It might well be that the intrinsic motional flexibility of α2 helix allows binding of bulky active site ligands. When we overlaied the crystal structures of rabbit ALOX15 (conformer A) with those of liganded human ALOX15B (Figure 1A) and with catalytic domain of the AA 12-lipoxygenating porcine ALOX15 (Figure 1B) we also observed a similar relative position of α2 helix. In previous studies we reported that tight association of the PLAT domain and the catalytic subunit of rabbit ALOX15 may be important for stability and integrity of α2 helix [20]. Hydrogen-deuterium exchange mass spectrometry on human ALOX15B also suggested that substrate binding might induce alterations in the structural dynamics of the PLAT domain and of α2 helix [21]. Thus, the conformation of α2 helix observed in conformer B of rabbit ALOX15 might not be considered as “non-native structure” but be a consequence of the high degree of structural flexibility of the catalytic domain.

To explore the conformational dynamics of ALOX15 monomers and the possibility of their functional cooperation in more detail we analyzed highly purified rabbit ALOX15 and human ALOX15B by native polyacrylamide gel electrophoresis (PAGE). When these enzyme preparations, which showed a single protein band in SDS-PAGE, were analyzed under native conditions we detected two well-separated protein bands and these data suggested structural heterogeneity of the enzyme preparations. When we mutated key amino acids that contribute to the inter-monomer interaction, we observed alterations in the catalytic properties and the mechanistic basis for the observed alterations was explored by molecular dynamics (MD) simulations.

## 2. Results

### 2.1. Structural Characterization of ALOX15 Isoenzymes by Native PAGE

To explore the structural heterogeneity of different ALOX-isoforms in aqueous solutions we selected two ALOX isoenzymes (rabbit ALOX15 and human ALOX15B), for which crystal structures have been solved (PDB entries 2P0M and 4NRE, respectively) [14,22]. We expressed the enzymes in *E. coli*, purified them to apparent homogeneity (>98% in SDS-PAGE) using consecutive affinity chromatography on Ni-NTA resign, anion-exchange and size-exclusion chromatography. Finally, we analyzed the purified proteins by different types of polyacrylamide electrophoresis (PAGE). The final preparations of the two enzymes migrated in SDS-PAGE (denaturing conditions) as single protein bands with an apparent molecular weight of 75 kDa (data not shown). The two enzymes also migrated as single bands in native PAGE (Figure 1C-I) when the enzymes were kept in 20 mM Tris-HCl buffer containing 130 mM NaCl (near physiological conditions). This data suggests structural homogeneity of the two ALOX-isoforms under these experimental conditions. However, when the purified protein was kept in the absence of salt (20 mM Tris-HCl) and subsequently analyzed by native PAGE two protein bands were observed (Figure 1C-II). It should be stressed at this point, that these enzyme preparations migrated as single band proteins when SDS was added to the running buffer (Figure 1C-III).

Taken together, these data suggested a structural heterogeneity (at least two different conformers) of both rabbit ALOX 15 and human ALOX15B in salt-free aqueous solutions. The different electrophoretic properties of the two conformers may either be related to a different surface charge, to a different overall shape or to a mixture of both. Although rabbit ALOX15 and human ALOX15B only share a moderate (37%) degree of amino acid identity, an overlay of their crystal structures shows a high degree of structure conservation (Figure 1A).

### 2.2. Consequences of N-terminal Truncation and Point Mutations of His585 on Conformational Heterogeneity of Rabbit ALOX15

In order to explore the mechanistic basis for the structural heterogeneity of rabbit ALOX15 (Figure 1C-II) we carried out three additional lines of experiments: (i) Since the N-terminal PLAT domain of this enzyme has been implicated in the structural hetero-geneity of rabbit ALOX15 [15,23], we prepared the catalytically active PLAT domain truncation mutant, purified the corresponding protein to apparent homogeneity and analyzed the final enzyme preparation by native PAGE. Here again (Figure 2A) we observed protein double bands for both, wild-type rabbit ALOX15 and its PLAT domain truncation mutant. This finding suggests that the structural heterogeneity of the two proteins, which is indicated by the protein band doublet, may not be related to the motional flexibility of N-terminal PLAT domain relative to the catalytic subunit. Thus, the structural heterogeneity is an intrinsic property of the catalytic domain. (ii) When we lowered the pH from 8.0 to 6.8, we also observed a protein band doublet but the relative intensities of the two bands were altered (Figure 2B). At pH 8.0 the two protein bands had similar intensities. In contrast, at pH 6.8 the trailing band was more intense. (iii) Finally, we carried out a number mutagenesis studies targeting amino acid residues that are of potential relevance for dimer formation [13]. According to the PISA analysis (https://www.ebi.ac.uk/msd-srv/prot_int/cgi-bin/piserver (accessed on 13 February 2021)) His585 (conformer A, helix α18) and Trp181 (conformer B, helix α2) contribute to the inter-monomer contact plane in the crystal structure of rabbit ALOX15 with their buried surface area of 86.65 and 153.82 Å^2^, respectively. The achieved CSS (complex formation significance score) implies that the interface plays an auxiliary role in complex formation.

In the monomeric state of both conformers Trp181 is solvent exposed (not shown). In contrast, His585 becomes solvent exposed only in the monomeric state of conformer A (Figure 2C), whereas in conformer B (Figure 2D) the side chain of this residue is shielded by α2 helix of the same conformer (Figure 2D). Here it apparently interacts with the side chain of Glu185 (α2 helix) (Figure 2D). Introduction of a negatively charged Glu at this position (His585Glu mutation) is likely to induce electrostatic repulsions between the negatively charged side chains of Glu185 and Glu585. These repulsive forces may impact the spatial arrangement of α2 and α18 helices. On the other hand, the negatively charged Glu585 might interact with the positively charged Lys189. These adhesive electrostatic forces may stabilize the structure of this conformer. Thus, it is hard to predict the functional consequences of such mutations. Similar scenarios were expected when a positively charged Lys is introduced at position 585. When His585 was mutated to either Glu or Lys we observed alterations in the retention properties of the proteins at the anion exchange matrix (Figure 2E). The His585Glu mutant was more strongly retained at the anion exchange column as indicated by the longer retention times. In contrast, the His585Lys mutant was early eluting, and these data suggest a weaker binding at the anion exchange matrix. Taken together, these data suggest that introduction of a negative charge at position 585 improved the enzyme–matrix interaction. In contrast, His585Lys exchanged impaired it (Figure 2E). From Figure 2F (left panel) it can be seen that the wild-type enzyme and both mutants (His585Lys and His585Glu) were co-eluted with the same elution volume for which a hydrodynamic radius of 30.5 ± 0.1 Å was estimated and these data are in accordance with our previous reports [15]. In hypotonic agueous solutions (20 mM Tris-HCl) at concentrations 10–15 µM or below rabbit ALOX15 is present predominantely as monomer [15]. Thus, the results obtained here suggest that under our experimental conditions the protein fractions (wild-type rabbit ALOX15 and the mutants) preapared by size exclusion chromatography (Figure 2F) also represent their monomeric forms.

Next, we analyzed these mutants by native PAGE (Figure 2G,H). For the His585Lys mutant we observed a reduced electrophoretic mobility when compared with the wild-type enzyme. At both pH values (8.0 and 6.8) the majority of the ALOX protein was trailing the wild-type enzyme (red-labelled enzyme pool III). These data indicate that the residues at position 585 contribute to the surface net charge of the protein. The His585Glu mutant was co-eluted with the leading band observed for the wild-type enzyme (red-labelled enzyme pool I).

To explore the impact of His585Glu exchange on the protein structure we quantified the intensities of the CD signals of wild-type rabbit ALOX15 at different temperatures with the corresponding values of the mutant enzyme. While the CD spectra of wild-type ALOX15 and its Trp181Glu mutant were almost identical, a deviation of the sigmal intensity at 220 nm was observed for the His585Glu mutantat at low temperatures (Figure 2I). Wild-type ALOX15 [20] and its His585Glu mutant undergo similar two-state unfolding transitions above 25 °C and these data suggest a similar thermostability of the two enzyme variants (Figure 2J). However, introduction of a negatively charged Glu alters the protein stability at lower temperatures. Thus, Glu585 may play a role for protein stability.

The fluorescence spectra of the wild-type ALOX15, its His585Glu, and its Trp181Glu mutants were very similar (Figure 2K) and these data suggest a similar tertiary structure of these three enzyme variants. Finally, we tested the response of wild-type human ALOX15 and its His585Glu mutant towards different guanidine hydrochloride (GdnHCl) concentrations (Figure 2L). GdnHCl molecules bind to peptide bonds leading to protein denaturation [24]. Solvent exposed bonds are more sensitive to GdnHCl denaturation. In a previous study [25] we have shown that the GdnHCl denaturation curve of rabbit ALOX15 follows a 3-state transition model (ground state M ↔ intermediate sate I ↔ unfolded state U), in which the M↔I transition was characterized by a high free energy (steep slope). Similar data were obtained for another wild-type enzyme preparation and the overall shape of the denaturation curve of the His585Glu mutant was very similar. However, between 1.0 and 1.3 M of GdnHCl, the slope of the denaturation curve of the mutant enzyme was steeper than that of the wild-type ALOX15 and these data suggested that less energy is required for the M↔I transition of the mutant enzyme (Figure 2L). In other words, the His585Glu mutant may be initially present in a more hydrated ground M state then the wild-type enzyme.

### 2.3. Impact of Point Mutations of Amino Acids Localized at the Inter-Monomer Contact Site on the Reaction Kinetics of Rabbit ALOX15

In aqueous solutions rabbit ALOX15 exists in a dynamic non-covalent monomer-dimer equilibrium and this steady state depends on protein concentration, absence and presence of salt and pH [15]. Dilution of the enzyme solution shifts this equilibrium towards ALOXmonomers and at concentrations below 10 µM dimers are virtually absent. However, in the crystal structure Leu179/Leu183 of the α2 helix of conformer A and Leu188/Leu192 of the α2 helix of conformer B form a hydrophobic leucin zipper, in which the leucine side chains of one monomer get in direct physical contact with the corresponding structural motif of the partner monomer. In addition, Trp181 of the α2 helix may interact with His585 of α18 helix (Figure 3A) to stabilize the dimer interface and contribute to the intermolecular communication. Previous mutagenesis studies combined with SAXS measurements suggested the importance of the Leu-cluster and of the site chains of Trp181 and His585 for ALOX15 dimerization [13]. In order to explore the contribution of individual amino acids to the allosteric character of ALOX15, wild-type rabbit ALOX15 and the corresponding single point mutants (Trp181Glu and His585Glu) were expressed, purified and their catalytic properties were assayed using arachidonic (AA) or linoleic (LA) acids as substrates.

As shown in Table 1, oxygenation of the two substrates follows Michaelis–Menten kinetics and K_M_ values of 21.4 ± 1.3 µM and 8.1 ± 0.4 µM for LA and AA, respectively, were calculated for rabbit ALOX15. The maximal catalytic activity for LA oxygenation (k_cat_) was four times higher than that of AA oxygenation, but the catalytic efficiencies (*k_cat_*/K_M_) of LA and AA oxygenation were very similar. This kinetic data is in line with previously published results [25]. Introduction of a negatively charged residue at position 585 (His585Glu) reduced catalytic efficiency of LA oxygenation but hardly altered the substrate affinity of the enzyme. For AA, the mutation induced a 2.5-fold increase in the catalytic efficiency making AA a better substrate than LA (Table 1, Figure 3C). Trp181Glu mutation had almost no effect on LA oxygenation parameters. In contrast, a 5-fold increase in the catalytic efficiency of AA oxygenation was observed. As we have previously reported, alterations in pH modifies the reaction specificity of arachidonic acid oxygenation by human and rabbit ALOX15 [26]. In contrast, salt and temperature alterations did hardly impact this enzyme property. In this study we have not explored whether the reaction kinetics are impacted by alterations in pH and salt concentration.

### 2.4. MD Simulation of the Structural Alterations of Rabbit ALOX15 Induced by Trp181Glu Exchange

To explore the structural basis for the observed functional alterations induced by amino acid exchanges at the inter-monomer interface we carried out MD simulations. Since Trp181Glu exchange induced the most pronounced functional alterations (5-fold increase in catalytic efficiency) we selected this mutant for our in-silico experiments. First, a model for the Trp181Glu mutant of the rabbit ALOX15 dimer was created on the basis of the X-ray coordinates of the wild-type enzyme. After the system was equilibrated, we analyzed the structural changes at the inter-monomer interface of the mutated dimer and compared it with the wild-type enzyme. The hydrophobic Leu zipper [Leu 179(A), Leu 183(A), Leu 188(B) and Leu 192(B)] at the inter-monomer interface observed in the crystal structure for the wild-type enzyme was largely preserved in the Trp181Glu mutant over the entire simulation period (100 ns) but the interface became more open in the mutant enzyme (Figure 4A,B). In fact, the distance between Leu179(A) and Leu192(B) was significantly increased and Leu183(A) and Leu188(B) did move apart. The Trp181(B)-His585(A) interaction was disrupted and Glu181(B) and His585(A) were re-oriented to form a very weak hydrogen bond. In the mutant enzyme, Glu181(A) and His585(B) were far apart (17 Å vs. 12 Å in the wild-type system). Another interesting structural alteration that was induced by Trp181Glu exchange was a rearrangement of the two α2 helixes (Figure 4B). The α2 helix of conformer A still involved seven turns, but it was somewhat bent. The α2 helix of conformer B underwent dramatic distortion, maintaining only two of its five turns (Figure 4B). The loss of the alpha-helix content contributes to the disruption of the hydrophobic cluster between the α2 helixes of the two monomers and to the loss of interactions between Glu181(B) and His585(A). In summary, our MD simulations of the Trp181Glu-ALOX15 mutant dimer suggested that the Trp181Glu dimer is structurally less stable.

### 2.5. AA and LA Binding Modes

To explore the substrate alignment of LA and AA at the active site of the Trp181Glu mutant we first carried out in silico docking studies. For this purpose, we placed the two fatty acids into the substrate binding pocket of conformer B. Next, we selected the most stable docking poses and run MD trajectories of 100 ns for both, the Trp181Glu-ALOX15-AA and the Trp181Glu-ALOX15-LA complexes. The most representative conformation of AA and LA bound to Trp181Glu-ALOX15 in the cavity of monomer B was obtained from clustering analysis. Those clusters were classified according to an RMSD of 0.5 Å based on all the heavy atoms of AA and LA, respectively. The centroid of the most populated cluster of each substrate along the MD simulation period was calculated. The structures of those centroids for AA and LA in the Trp181Glu-ALOX15 complex are depicted in Figure 4C,D, respectively. For comparison, the representative centroid structure of AA and LA binding modes, in the cavity of monomer B for the corresponding wild-type ALOX15 dimer, has also been plotted in Figure 4C,D. The MD simulations of the two wild-type substrate-bound dimers (ALOX15-AA and ALOX15-LA) were carried out previously [17]. The results for the Trp181Glu-ALOX15-AA system showed that AA is stretched out inside the substrate binding pocket of the Trp181Glu mutant occupying the whole available space. The AA hydrophobic tail is located near the bottom of the cavity surrounded by the triad determinants Ile418, Met419, and Ile593 [27]. During the whole MD simulation period, the carboxylate group of AAs was fixed by a hydrogen bond to Trp145 [O(AA)-H-Trp145 distance of 2.4 Å] at the centroid structure in Figure 4C. LA is also stretched out with its tail close to the bottom of the cavity.

The LA carboxylate group was immobilized during the MD simulation by three hydrogen bonds, as depicted in Figure 4D. One hydrogen bond was established with Arg405 [d(O(LA)- H(Arg405) = 2.204 Å], a second one with Asn152 [d(O(LA)- H(Arg405) = 3.346 Å] and a third one with Lys146 [d(O(LA)- H(Arg405)) = 1.872 Å]. Our MD simulations do not support a U-shaped binding mode [28] for AA in the Trp181Glu-ALOX15 dimer mutant. In contrast, the U-shaped model has previously been confirmed in simulations of the AA complex of the N-terminal truncation mutant of pig ALOX15 [29] and of the corresponding complex of AA with coral 8R-LOX [30].

### 2.6. Analysis of Pre-Catalytic Structures

In Figure 4E the evolution of the distance between C_13_ of AA and the oxygen atom of the OH-group of the Fe^3+^-OH cofactor is shown along the MD trajectory for the Trp181Glu-ALOX15-AA system. The corresponding distance in the WT-ALOX15-AA complex simulation [17] is included for comparison. When bound in the cavity of conformer B of the Trp181Glu-ALOX15-AA system, C_13_ of AA approaches the Fe^3+^-OH cofactor in a similar way as we observed it for the WT-ALOX15-AA dimer (Figure 4E), but the fluctuations of the C_13_-OH distance during the whole MD simulation were less pronounced for the Trp181Glu mutant. In Table 2 the average C_13_-OH, H_13proS_-OH and H_13proR_-OH distances for AA are given for the Trp181Glu-ALOX15-AA and WT-ALOX15-AA complexes. The average C_13_-OH distance is 1.5 Å shorter in the Trp181Glu mutant when compared with the wild-type enzyme. As shown in previous studies [31,32], the pre-catalytic structures for the catalytic mechanism of lipoxygenases have been defined according to two criteria d(HX-OH) ≤ 3 Å and d(HX−OH) < d(CX-OH) (X = 13). The second condition ensures that the corresponding CX−HX bond is properly oriented for hydrogen abstraction. This criterion has been used to filter out “well-oriented” structures. In the present study, a very high percentage of well-oriented structures was observed. In previous studies we have shown that owing to the structural fluctuations of AA along the MD trajectory in the WT-ALOX15-AA complex, the average C_13_-OH distance was rather long. In contrast, in the Trp181Glu-ALOX15-AA complex AA remains more rigidly bound in the substrate binding pocket so that the average C_13_-OH distance became shorter. Consequently, the percentage of pre-catalytic structures significantly increases by a factor of 16 in the case of the mutant.

Figure 4F presents the evolution of the distance between C_11_ of LA and the oxygen atom of the iron-bound OH-group along the MD trajectory for the Trp181Glu-ALOX15-LA system. The corresponding distances in the WT-ALOX15-LA complex simulation are included for comparison. When bound in the cavity of monomer B of the Trp181Glu-ALOX15-LA system, C_11_ of LA fluctuates around an average distance to the Fe^3+^-OH cofactor in a similar way as we observed it during the WT-ALOX15-LA complex simulation (Figure 4F). Only during the last 25 ns, the fluctuations showed significant differences. Table 3 summarizes the average C_11_-OH, H_11proS_-OH, and H_11proR_-OH distances for LA obtained for the Trp181Glu-ALOX15-LA and WT-ALOX15-LA complexes. The average C_11_-OH LA distance is only 0.3 Å shorter in theTrp181Glu mutant when compared with the WT system. However, it is longer than the C_13_-OH distance for AA in the same system. The percentage of well-oriented structures for LA is somewhat lower in the Trp181Glu-ALOX15-LA complex when compared with the WT system. As for the pre-catalytic structures, we obtain a 1.6-fold increase in the case of the mutant, which is substantially smaller than that obtained for AA.

The results obtained during the MD simulations of the Trp181Glu-ALOX15-AA and WT-ALOX15-LA dimer complexes suggest that AA is bound at the active site of the mutant enzyme in a such way that less energy would be needed to catalyze hydrogen abstraction from C_13_ (Table 2). This simulation data is consistent with the 5-fold increase in the catalytic efficiency observed for AA oxygenation in the Trp181Glu system (Table 1). On the other hand, for the LA complexes much smaller differences were calculated (Table 3) when wild-type ALOX15 and its Trp181Glu mutant were compared. This finding is in accordance with our experimental data (Table 1).

## 3. Discussion

### 3.1. Conformational Heterogeneity of ALOX15

Mammalian ALOX-isoforms constitute single polypeptide chain proteins, which fold into a two-domain structure [33,34]. The large C-terminal catalytic domain consisting of several alpha helices involves the fatty acid binding cavity and the catalytic non-heme iron. The N-terminal PLAT domain has been implicated in activity regulation and membrane binding [23]. In 2008 Choi et al. [14] re-evaluated the set of X-ray coordinates deposited in PDB for a rabbit ALOX15-inhibitor complex [33] and found that the enzyme is present in the crystals in two structurally distinct conformers. Conformer A involves an empty substrate binding pocket but conformer B carries the RS7 inhibitor inside this cavity. In addition to this difference, there are a number of other distinct structural characteristics, in particular the different orientations of α2 and α18 helices. According to the re-interpreted structural model (PDB 2P0M) rabbit ALOX15 is present as a mixed protein dimer, in which the two conformers A and B interact with each other via their α2 and α18 helices. In the crystals the spatial arrangement of the different secondary structural elements of conformers A and B are very similar except for those helices (α2 and α18 helices) that directly interact with each other within the dimer. If one overlays the structures of conformer A (empty substrate binding pocket) with that of conformer B (ligand present in the substrate binding pocket) one can conclude that ligand binding in the substrate binding pocket induces two major structural alterations: (i) α18 Helix undergoes minor rearrangement with respect to its relative orientation to other helices. (ii) α2 Helix is displaced by about 12 Å and is shortened. In fact, the helical character is lost for amino acids 169–176 (Figure 3B). However, it is not clear if the ligand binding at the active site induces alterations in the overall protein structure. Previous investigations suggest that the spatial orientation of the α2 helix is important for the non-covalent interaction of the C-terminal catalytic domain with the N-terminal PLAT domain, but also for the inter-monomer interface within the ALOX15 dimer [17,20].

Here we report that recombinant rabbit ALOX15 as well as recombinant human ALOX15B is present in aqueous solutions in two conformers, which can be separated by native PAGE (Figure 1C). The observation that our enzyme preparation, which migrates as a single protein band in SDS-PAGE, is heterogenous (two protein bands) in native PAGE (Figure 2A,B) suggests that the enzyme is present in two different conformers, which can be separated from each other on the basis of their distinct electrophoretic mobilities under native conditions. In a previous study [15] we have shown that pure rabbit ALOX15 is present in aqueous solutions as a reversible monomer-dimer equilibrium. This equilibrium depends on the protein concentration, presence and absence of salt and on pH [15]. Dilution of the enzyme solution strongly favor monomer formation. In the present study the enzyme concentration is below 10 µM. Under these experimental conditions monomers strongly prevail and dimers are virtually absent. Thus, the two bands observed in native PAGE represent different monomeric enzyme conformers but not a monomer-dimer mixture. To further support this conclusion, we performed size exclusion chromatography with our enzyme preparations (Figure 2F, left panel). Here we observed identical retention volumes and the estimated hydrodynamic radius of the wild-type protein is identical with the corresponding values of the mutant enzymes. Here again, no protein dimers were observed, which should be present as front shoulders of the monomer peak.

Our mutagenesis studies combined with size exclusion chromatography and electrophoretic mobility assays suggested that the conformational heterogeneity is an intrinsic property of the catalytic domain. However, it remains unclear whether the two bands we observed in native PAGE may correspond to the two conformers observed in the crystal structure.

### 3.2. Cooperativity of Rabbit ALOX15 Monomers during Lipoxygenase Reaction

Previous mutagenesis studies combined with SAXS measurements [13] suggested the importance of a hydrophobic Leu-zipper and of Trp181/His585 interaction (Trp181 belongs to the α2 helix of conformer B and His585 to helix α18 of conformer A) for ALOX15 dimerization (Figure 3A). Mutations of these hydrophobic amino acid to negatively charged Glu hinders protein dimerization [13]. Here we found that introduction of negative charges at positions 181 or 585 (Trp181Glu and His585Glu) reduced the catalytic efficiency of LA oxygenation. In contrast, this readout parameter was significantly elevated for AA oxygenation (Table 1, Figure 3C). It is difficult to interpret the structural basis for the observed functional alterations summarized in Table 1. Since His585 and Trp181 do not directly contact any residues of the substrate binding pocket these mutations should not directly modify the structure of the active site. In fact, no major differences were observed between wild-type ALOX15 and the corresponding enzyme mutants when their fluorescence spectra (tertiary structure) were compared (Figure 2K). Thus, from the structural point of view, these mutations should not dramatically alter the reaction kinetics of fatty acid oxygenation. However, we did observe significant kinetic differences between the wild-type enzyme and the His585Glu and the Trp181Glu mutants and these data suggest that the mutations may impact the catalytic properties of the enzyme in an indirect way. Since His585 and Trp181 are located at the inter-monomer interface we hypothesized that the amino acid exchanges may modify the inter-monomer interactions. These modifications may be considered as the structural basis for the observed functional alterations.

As we have shown in our previous works [31,32] the spatial orientation of hydrogens attached to the bisallylic methylene groups (C_11_ for LA and C_13_ for AA) relative to the Fe^3+^-OH cofactor is important for the catalytic mechanism of the lipoxygenase reaction. The length of the aliphatic hydrocarbon chain, the number and position of the double bonds as well as the geometry of the double bond system may impact the alignment of fatty acid substrates at the active site and makes LA a better substrate than AA for rabbit wild-type ALOX15. In contrast, AA was a better substrate for the Trp181Glu mutant. A similar trend was observed for the His585Glu mutant. His585 is a constituent of the α18 helix. This structural motif limits the deepness of the substrate-binding pocket [17,27], and thus should be important for substrate binding. On the other hand, Trp181 is localized in far distance to the substrate-binding pocket and there is no immediate connection of this amino acid to the active site. Thus, it remains unclear how the Trp181Glu exchange is translated into the observed kinetic differences. It might be possible that the allosteric character of the enzyme is involved but there are certainly alternative explanations.

In our previous 4 ns MD simulations of the His585Glu+Trp181Glu double mutant we observed that this mutation induced structural alterations of the inter-monomer interface [13]. In the present study we carried out 100 ns MD simulations using improved force field characteristics. Here we found that the Trp181Glu exchange did not completely disrupt the inter-monomer interface but led to significant modification of the α2 helix. While the α2 helix of conformer A retains its seven turns during MD simulation, the α2 helix of conformer B loses three of the five turns observed in the x-ray structure. Such dramatic structural differences in the inter-domain interface structure may further be translated into the catalytic center. In fact, our MD simulations suggested that AA is bound at the active site of the mutant enzyme in a such way that less energy would be required for abstraction of hydrogens attached to C_13_ of AA. In the Trp181Glu-ALOX15-AA complex the substrate is more rigidly bound in the substrate binding pocket so that the fluctuations of the C_13_-OH distances during the whole MD simulation are less pronounced and the average C_13_-OH distance became shorter.

## 4. Materials and Methods

### 4.1. Chemicals

The chemicals used were obtained from the following sources: arachidonic acid (5Z,8Z,11Z,14Z-eicosatetraenoic acid) from Cayman Chem (distributed by Biomol, Hamburg, Germany), guanidinium hydrochloride (GdHCl) from Sigma-Aldrich (St. Louis, MO, USA), HPLC grade methanol, acetonitrile, and acetic acid from Fisher Scientific (Nidderau, Germany), isopropyl-β-d-thiogalactopyranoside (IPTG) from Carl Roth GmbH (Karlsruhe, Germany). The *E. coli* strain XL-1 blue was purchased from Stratagene (La Jolla, CA, USA), the Rosetta 2 strain BL21(DE3)pLysS was purchased from Invitrogen (Carlsbad, CA, USA).

### 4.2. Expression of Rabbit ALOX15

Wild-type rabbit ALOX15 and the enzyme mutants were expressed as N-terminal His-tag fusion proteins in *E. coli* using the pQE-9 prokaryotic expression plasmid and XL-1 blue competent cells according to the protocol described previously [20].

### 4.3. Expression of Human ALOX15B

For this study, the human ALOX15B cDNA was cloned into the pQE-9 expression plasmid (Qiagen, Hilden, Germany) between the BamH I and Hind III restriction sites. The starting Met was mutated to a Val and a Hind III restriction site was introduced immediately behind the stop codon. The N-terminus of the recombinant his-tag fusion human ALOX15B reads MRGSHHHHHHGSSV- ALOX15B. The original starting Met of the enzyme was exchanged to a Val. The recombinant enzyme was expressed in *E. coli* (XL-1 Blue cells). Bacteria were transformed with the recombinant plasmid and 1.5 L LB media containing 100 mg/L ampicillin were inoculated with a 20 mL overnight pre-culture. Bacteria were allowed to grow at 37 °C for 16 h. After addition of 1 mM isopropyl-1-thio-β-D-galactopyranoside (IPTG) the cultures were kept for additional 3 h at 28 °C. Then, bacteria were spun down, washed with PBS and resuspended in 100 mL of PBS. Cells were disrupted at 15,000 psi with an Emulsiflex-C5 high-pressure cell homogenizer (Avestin, Ottawa, ON, Canada) and cell debris was removed by centrifugation (30 min, 14,000 g).

### 4.4. Purification of Rabbit ALOX15 and Human ALOX15B

The lysis supernatants were incubated with 1.5 mL of Ni-NTA resin (Invitrogen, Toulouse, France) for 1 h at 4 °C. The resin with the bound protein was transferred to an open bed 10-mL column. The column was washed two times with 2 mL washing buffer I (100 mM Tris, pH 8.0, 200 mM NaCl, 10 mM imidazole) and two times with 2 mL of washing buffer II (100 mM Tris, pH 8.0, 200 mM NaCl, 25 mM imidazole). Finally, the adhering proteins were eluted five times with 0.6 mL elution buffer (100 mM Tris, pH 8.0, 200 mM NaCl, 200 mM imidazole). Enzyme containing fractions were combined, the enzyme solution was desalted using an Econo-Pac 10DG column (Bio Rad, Munich, Germany), and the ALOX-isoforms were further purified by conventional anion exchange chromatography. For this purpose, an FPLC-system (GE Healthcare Bio-Sciences AB, Uppsala, Sweden) equipped with a Resource Q 6-mL column was used and adhering proteins were eluted by a linear NaCl gradient. 20 mM Tris (pH 6.8 or 8.0) was used as elution buffer A and 20 mM Tris/1M NaCl (pH 6.8 or 8.0) as buffer B. The fractions containing active ALOX protein were pooled and homogeneity of the samples was checked by size exclusion chromatography. For this purpose, the samples from the previous step were run on a Superdex™200 10/300 GL column (GE Healthcare Bio-Sciences AB, Uppsala, Sweden) using 20 mM Tris–HCl buffer (pH 6.8 or 8.0) at a flow rate of 0.5 mL/min. For each run a solution of 75–100 µg of ALOX protein in 100 µL of was applied (final concentration 10–13 µM).

### 4.5. Estimation of the Hydrodynamic Radii for Wild-Type and Mutant ALOX15

Prior to analysis the column (Superdex™200 10/300 GL column, GE Healthcare Bio-Sciences AB, Uppsala, Sweden) was calibrated using chymotrypsinogen A (20.9 Å), bovine serum albumin (35.5 Å) and catalase (52.2 Å) as size standards (Figure 2F, right panel) for estimation of the hydrodynamic radii (R_h_). Dextran blue 2000 was used to estimate the column void volume. The K_av_ values were calculated using equation K_av_ = (V_e_ − V_o_)/(V_t_ − V_o_), where V_e_ is the elution volume of each protein, V_o_ the void volume of the column and V_t_ the total bed volume of the column. The (−logK_av_)^1/2^ values for standard proteins were plotted vs. corresponding hydrodynamic radius (R_h_).

### 4.6. Native PAGE Electrophoresis

To study conformational heterogeneity of the purified ALOX15 proteins gradient Mini-PROTEAN^®^ TGX^TM^ precast gels 4–15% (Bio Rad, Feldkirchen, Germany) were used. The samples were freshly prepared and maintained in 20 mM Tris-HCl buffer, pH 8.0 or pH 6.8, containing either no salt or 130 mM NaCl. Each sample (10 µl) contained 300–400 ng (final concentration 0.4–0.6 µM) ALOX protein. A 4-fold concentrated loading buffer (400 mM Tris-HCl, pH 8.0 or 6.8, 40% glycerol and 0.01 mL bromophenol blue were added. The running buffer consisted of 25 mM Tris-HCl buffer containing 192 mM glycine. Native PAGE was carried out at a constant voltage of 100 V at room temperature for 180 min. The gels were stained with a standard silver-staining protocol.

### 4.7. CD Measurements

CD measurements were performed using a Jasco-710 spectropolarimeter. Spectra were collected in the range 200–250 nm with a 0.1 cm path length quarts cell. For all experiments, the concentration of the samples was 0.13 mg/mL and the temperature was maintained constant by a circulating water bath. The temperature range used was between 10 to 60 °C.

### 4.8. Guanidine Denaturation Studies.

Guanidine denaturation studies were performed as described previously [25]. Briefly, the protein was incubated in the presence of different amounts of guanidine hydrochloride for 12 h at 4 °C. Each measurement was repeated at least three times, and the relative standard deviation was reported in the figures as error bars. Steady-state fluorescence spectra were recorded with a ISS-K2 fluorimeter (ISS, Champaign, IL, USA) using an excitation wavelength of 280 nm.

### 4.9. Rabbit ALOX15 Activity Assay (Purified Enzyme)

The oxygenation kinetics of linoleic acid [LA] or arachidonic acid [AA] were assayed spectrophotometrically measuring the increase in absorbance at 235 nm at different substrate concentrations. The assay mixture was composed of substrate solution (applied as sodium salt) in 1 mL PBS and the reaction was started by the addition of enzyme (2.5 µg, specific activity with LA - 30 s^−1^). The linear parts of the kinetic progress curves were evaluated. All measurements were carried out at the room temperature in triplicate. For each enzyme preparation the amounts of protein added to the assay mixture was normalized to a 1:1 stoichiometry of apoenzyme: non-heme iron.

### 4.10. Molecular Docking Simulations

Docking calculations were carried out with the program GOLD5.8.0 [35]. Hydrogen coordinates were generated with the H++ web-server [36,37] using a pH = 6.0 for titratable residues. For AA and LA the docking calculations were performed fitting the substrate fatty acids into the substrate binding pocket of a relaxed structure of conformer B of the Trp181Glu-ALOX15 dimer. For this purpose, the active site ligand that occupies the substrate-binding pocket of conformer B was removed. The substrate-binding cavity used in our docking studies was a 20 Å radius sphere centered around the catalytic non-heme iron of conformer B. The enzyme structure was kept unaltered during the docking studies, but complete flexibility of the fatty acid substrates was permitted. We activated the option in GOLD that considers the interactions of organic ligands with metal ions in metalloenzymes but limiting the docking exploration to hexacoordinated geometries of iron. The most efficient genetic algorithm was used to ensure an extensive search of the conformational space of all substrates. To estimate the binding free energies of substrates the ChemScore fitness function was selected.

### 4.11. Molecular Dynamics (MD) Simulations

First, our setup used the W181E-ALOX15 dimer model as well as the W181E-ALOX15-AA and the W181E-ALOX15-LA dimer complexes using the protocol established by the AMBER program package [38]. The ff14SB [39] force field was used for the protein. The force field parameters for iron and its ligands [17] (His361, His366, His541, His545, Ile 663, and OH^−^) as well as for the two substrates (AA [40], and LA [41]) were adopted from previous studies using Antechamber and Parmchk2 modules following the protocol in AMBER. The GAFF2 [39,42] library was taken as source of those parameters. The carboxylate groups of the two substrates were modelled unprotonated to match the protonation state under physiological conditions. Using the tLeap program the different complexes were solvated with an orthorhombic box of pre-equilibrated TIP3P [43] waters and the total charge was neutralized with sodium cations. The final systems contain nearly 200,000 atoms with about 21,000 of them belonging to the protein. The rest of atoms were water molecules and salt ions. All molecular dynamics (MD) simulations were run with the AMBER 18 GPU (CUDA) version of the PMEMD package [44,45]. First, the systems were minimized using the steepest-descent and conjugate gradient algorithms for 22,000 energy minimization steps to avoid close contacts. Harmonic restraints were applied to the protein and substrate atoms with a force constant of 5.0 kcal mol^−1^ Å^−2^ in the first 6,000 steps. In the following 6,000 steps, harmonic restraints were applied only to the protein backbone and the substrate heavy atoms with a force constant of 5.0 kcal mol^−1^ Å^–2^. In the last 10,000 steps the whole system was free of restraints. Then, MD simulations using periodic boundary conditions were performed. The system was gradually heated from 0 K to 300 K for a period of 200 ps. Next, a MD run of 1 ns, at constant temperature and pressure (300 K, 1 bar), has been calculated to adjust the volume of the box to reach a density of around 1 g cm^−3^. Harmonic restraints were applied to the protein backbone and substrate heavy atoms with a force constant of 5.0 kcal mol^−1^ Å^–2^ during the heating and compressing steps. The temperature was controlled by Langevin dynamics [46], while the pressure was adjusted by the Berendsen barostat [47]. Then, an equilibration stage of 10 ns, at constant temperature (300 K) and volume, was carried out. Finally, a production period of 100 ns was run within the same isothermal-isochoric ensemble. A time step of 2 fs was used along the whole MD trajectory. All bonds and bends containing hydrogen atoms were constrained by the SHAKE algorithm [48]. Non-bonding interactions have been calculated with a cut-off of 9 Å. The last structure of the MD simulation for the Trp181Glu-ALOX15 dimer was then taken as receptor for docking AA and LA into monomer B. Next, the best docking pose of AA was selected as the starting structure for the MD simulations of Trp181Glu-ALOX15-AA, and the best docking pose of LA was taken to initiate the simulation of the Trp181Glu-ALOX15-LA- complex. For comparison, we have also included the results of the MD simulation of the WT ALOX15-AA and WT ALOX15-LA dimer complexes from our previous study [17]. Analysis of the MD simulations was carried out with AmberTools18, whereas visualization of those trajectories was performed with VMD [49] and USCF CHIMERA [50] programs.

## 5. Conclusions

Pure preparations of recombinant rabbit ALOX15 and human ALOX15B, which run as single band in SDS-PAGE, show structural heterogeneity when analyzed by native PAGE. These data suggest that ALOX15 is present in aqueous solutions in different conformers. Site directed mutagenesis of amino acids, which are important for the stability of the inter-monomer interface, induced structural alterations of the native dimer complex. These alterations are translated to the active site changing the catalytic activity of the enzyme.

## Figures and Tables

**Figure 1 ijms-22-03285-f001:**
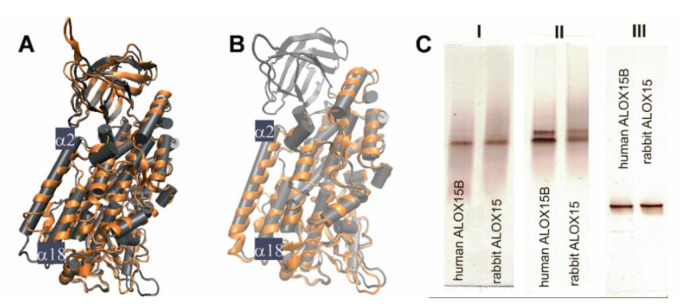
(**A**) Overlay of crystal structures of rabbit ALOX15 (PDB entry 2P0M, conformer A, grey) and human ALOX15B (PDB entry 4NRE, brown). (**B**) Overlay of crystal structures of rabbit ALOX15 (PDB entry 2P0M, conformer A, grey) and the AA 12-lipoxygenating porcine ALOX15 catalytic domain (PDB entry 3RDE, brown). (**C**) Structural heterogeneity of rabbit ALOX15 and human ALOX15 B in salt-free aqueous solutions. (C-I) Native polyacrylamide gel electrophoresis (PAGE). For this analysis, the enzymes were prepared and maintained in 20 mM Tris-HCl buffer, pH 8.0 containing 130 mM NaCl. (C-II) Native PAGE. For this analysis, the enzyme was prepared and maintained in 20 mM Tris-HCl buffer, pH 8.0 that does not contain NaCl (desalted enzyme preparation eluted as a single peak by and size-exclusion chromatography). (C-III) Denaturing PAGE. For this analysis the enzyme was prepared and maintained in 20 mM Tris-HCl buffer, pH 8.0, but the running buffer contained 0.5% SDS.

**Figure 2 ijms-22-03285-f002:**
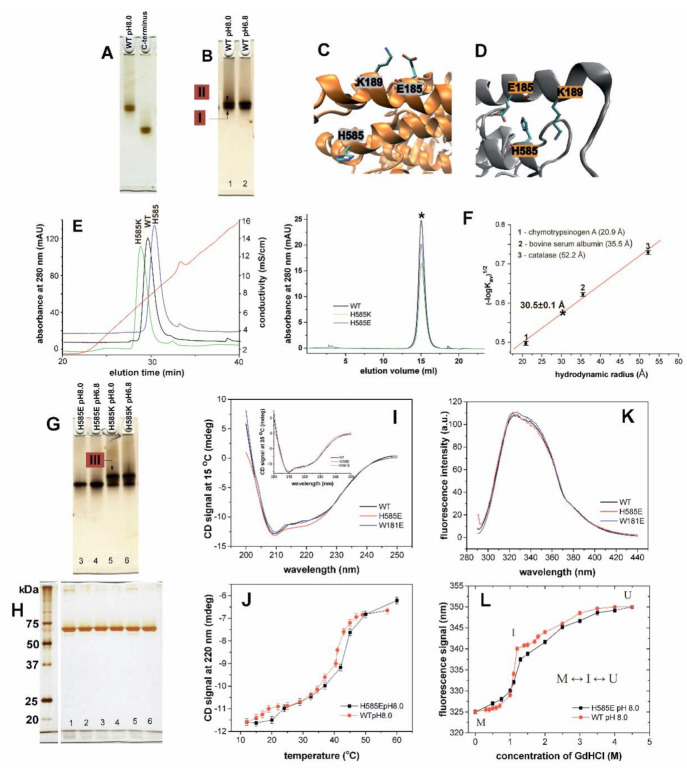
Structural characterization of wild-type and mutant rabbit ALOX15. (**A**) Native PAGE electrophoresis of rabbit ALOX15 and its catalytic domain. (**B**) Effect of pH on structural heterogeneity of ALOX15. At pH 8.0 the two conformers (red labelled enzyme pools I and II) are better resolved when compared with pH 6.8. (**C**) Crystal structure of conformer A (no ligand at the active site). In this structure the side chain of His585 (H585) is localized on the protein surface and is accessible to the solvent. (**D**) Crystal structure of conformer B (ligand bound at the active site). Here the side chain of His585 (H585) is buried inside the protein between the side chains of Glu185 (E585) and Lys189 (K189). It is shielded from the solvent. (**E**) Elution order of the purified ALOX15 variants from the Resource Q (6 mL) column in anion exchange chromatography using a linear NaCl gradient. Absorbance of the column effluent at 280 nm (green, black, blue) and conductivity (red curve) were simultaneously recorded. (**F**) Size exclusion chromatography (left panel) of the purified recombinant proteins was carried in 20 mM Tris-HCl buffer and we estimated their hydrodynamic radii using different calibration proteins (right panel). The elution volume of the wild-type ALOX15 and the mutants is labelled with asterisk. (**G**) Native PAGE of rabbit ALOX15 mutants was performed at two different pH as described above for the wild-type ALOX15. (**H**) SDS-PAGE of the enzyme preparations (the numbers correspond to the protein samples that are present on panels B and G). (**I**) CD spectra of wild-type and mutant rabbit ALOX15. (**J**) Thermal stability of wild-type rabbit ALOX15 and its His585Glu (H585E) mutant. The relative intensity of the CD signal of ALOX solution was measured at 220 nm at different temperatures. (**K**) Fluorescence spectra of wild-type and mutant rabbit ALOX15. (**L**) The shift of the maximum of the fluorescence spectrum in the presence of different concentrations of GdnHCl was monitored by fluorescence steady state spectroscopy for wild-type rabbit ALOX15 and its His585Glu (H585E) mutant. The GdnHCl denaturation curves follow a 3-state transition model (ground state M ↔ intermediate sate I ↔ unfolded state U).

**Figure 3 ijms-22-03285-f003:**
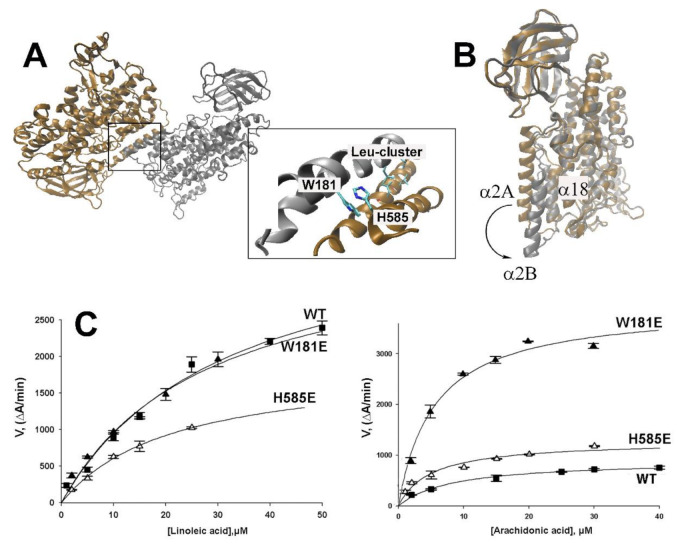
Contribution of Trp181 (W181) and His585 (H585) to the inter-monomer interface of rabbit ALOX15 and impact of mutations on the reaction kinetics of the enzyme. (**A**) Crystal structure of the rabbit ALOX15 heterodimer (PDB entry 2P0M) consisting of a ligand-free conformer A (brown) and a ligand-bound conformer B (grey). Inset: Amino acid residues contributing to the inter-monomer interface. (**B**) The two structures of the rabbit ALOX15. α2 Helix of conformer A (α2A) is strongly dislocated when compared with conformer B (α2B). (**C**) Reaction kinetics of wild-type rabbit ALOX15 and of two enzyme mutants. Linoleic (left panel) or arachidonic acid (right panel) oxygenation was assayed spectrophotometrically (increase in absorbance at 235 nm) at different substrate concentrations. For each measurement 56 nM LOX (final enzyme concentration) normalized to the iron content was used.

**Figure 4 ijms-22-03285-f004:**
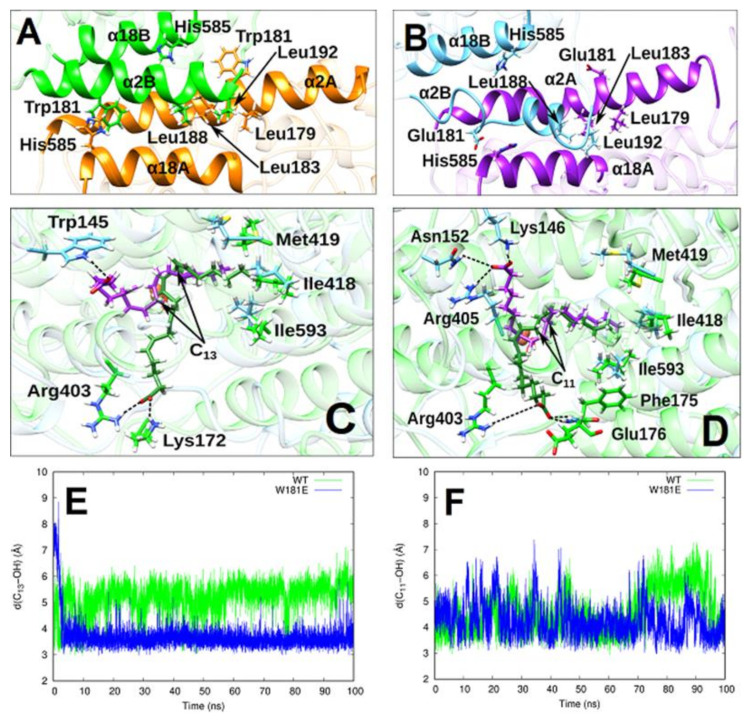
Structural consequences of Trp181Glu exchange on dimer formation and substrate alignment. (**A**) Inter-monomer interface of wild-type rabbit ALOX15. Secondary structural elements of conformer A are shown in mustard and those of conformer B are given in green. (**B**). Inter-monomer interface of the Trp181Glu mutant of rabbit ALOX15. Secondary structural elements of conformer A are shown in light blue and those of conformer B in purple. (**C**) Most representative binding mode of AA in WT-ALOX15 (dark green) and in Trp181Glu-ALOX15 (purple) dimers. WT-ALOX15 (green with a percentage of transparency) and Trp181Glu-ALOX15 (light blue with a percentage of transparency) backbones have been superimposed. The side-chains of some selected residues for WT-ALOX15 and Trp181Glu-ALOX15 dimers have been displayed in green and light blue, respectively. (**D**) Most representative binding mode of LA in WT-ALOX15 (dark green) and in Trp181Glu-ALOX15 (purple) dimers. WT-ALOX15 (green with a percentage of transparency) and Trp181Glu-ALOX15 (light blue with a percentage of transparency) backbones have been superimposed. The side-chains of some selected residues for WT-ALOX15 and Trp181Glu-ALOX15 dimers have been displayed in green and light blue, respectively. (**E**) C_13_-OH distance in the Trp181Glu-ALOX15-AA (in blue) and in the WT-ALOX15-AA (in green) complexes versus time. (**F**) C_11_-OH distance in the Trp181Glu-ALOX15-LA (in blue) and in the WT-ALOX15-LA (in green) complexes versus time.

**Table 1 ijms-22-03285-t001:** Catalytic activities of rabbit wild-type ALOX15 and selected mutants.

Enzyme	LA			AA		
	*k_ca_*_t_, s^−1^	K_M,_ µM	*k_cat_/*K_M,_ s^−1^µM^−1^	*k_ca_*_t_, s^−1^	K_M,_ µM	*k_cat_*/K_M,_ s^−1^µM^−1^
WT	47.2 ± 2.8	21.4 ± 1.3	2.2 ± 0.1	11.3 ± 0. 6	8.1 ± 0.4	1.4 ± 0.2
His585Glu	21.2 ± 3.1	19.8 ± 2.9	1.1 ± 0.2	15.8 ± 0.9	4.9 ± 0.3	3.2 ± 0.1
Trp181Glu	39.8 ± 6.2	24.7 ± 3.9	1.6 ± 0.2	54.40 ± 2.01	6.6 ± 0.4	8.3 ± 0.51

**Table 2 ijms-22-03285-t002:** Average distances of C_13_ (d(C_13_-OH)) and its hydrogen atoms H_13proS_ (d(H_13proS_-OH)) and H_13proR_ (d(H_13proR_-OH)) of AA to the iron-bound OH-group for the Trp181Glu-ALOX15-AA and WT-ALOX15-AA complexes ^1^.

System	d(C_13_-OH) (Å)	d(H_13proS_-OH) (Å)	d(H_13proR_-OH) (Å)	Well-Oriented Structures (%)	Pre-Catalytic Structures With At least One H Well- Oriented (%)
Trp181Glu	3.70	4.50	3.15	99.10	55.69
WT	5.19	4.85	5.47	97.78	3.40

^1^ The percentage of well-oriented structures and the percentage of pre-catalytic structures with at least one hydrogen atom well-oriented for hydrogen abstraction are also given.

**Table 3 ijms-22-03285-t003:** Average distances of C_11_ (d(C_11_-OH)) and its hydrogen atoms H_11proS_ (d(H_11proS_-OH)) and H_11proR_ (d(H_11proR_-OH)) of LA to the iron-bound OH-group for the Trp181Glu-ALOX15-LA and WT-ALOX15-LA complexes ^1^.

System	d(C_11_-OH) (Å)	d(H_11proS_-OH) (Å)	d(H_11proR_-OH) (Å)	Well-Oriented Structures (%)	Pre-Catalytic Structures With At least One H Well- Oriented (%)
Trp181Glu	4.26	4.17	4.18	87.99	22.65
WT	4.58	4.74	4.56	90.09	14.40

^1^ The percentage of well-oriented structures and the percentage of pre-catalytic structures with at least one hydrogen atom well-oriented for the hydrogen abstraction are also given.

## Data Availability

The data presented in this study is contained within the article.

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
