# Peer review of "Conformational Heterogeneity and Cooperative Effects of Mammalian ALOX15"

_ijms, 2021, doi:10.3390/ijms22063285_

Round 1

Reviewer 1 Report

Ivanov et al. observed the dimeric formation of purified rabbit and human ALOX15 in native gel. Based on the previously reported crystal structure, biochemistry was carried out through mutagenesis of important amino acids of α2 and α18 helices, which are considered to be involved in dimeric formation or stability in  ALOX15. Also, MD simulation was performed. I agree that these functional studies support that ALOX15 will act as an allosteric regulator with dimeric formation. Although this study showed functional differences for certain amino acids in α2 and α18 helices, I think that their results can be inferred from previously reported crystal structure and biochemical experiments. Therefore, although I think this manuscript is well written and believe scientifically for the results of the studies conducted by the authors, I believe that this study is at the level of supporting previous studies rather than providing new meaningful information. Meanwhile, I propose the following contents to improve the paper.

  1. Under salt-free conditions, Rabbit ALOX15 and human ALOX15B preferred dimers in native gel. On the other hand, it is judged that a weak dimer size was observed even in the condition of salt in the native gel. In protein purification, a buffer without salt was used in gel filtration. Was a dimer observed in this case? I believe that adding the gel filtration results under the same experimental conditions of the native gel can also increase the reliability of the results.
  2. In Figure or Table caption, some contents must be moved to the Method section or deleted.
    -Figure 1 caption:
    ”Rabbit ALOX15 and human ALOX15B were expressed in coli, purified by consecutive 121 affinity chromatography on Ni-NTA resign, anion exchange and size exclusion chromatography 122 and analyzed by PAGE under native and denaturing conditions.”
    “ 200-400 ng protein were applied to precast PAGE gradient gels (4-15%, Bio-Rad, Feldkirchen, Germany). The gel was run for 180 min.”
    - Figure 2 caption:
    “ 300 ng pure ALOX protein were applied to precast native PAGE gels, runtime 180 min.”
    “The samples were freshly prepared and maintained in 20 mM Tris-HCl buffer, pH 8.0 (see Methods and Materials).”
    “300 ng pure ALOX15 protein were applied to precast native PAGE gels, runtime was 180 min.”
    “Absorbance of the column effluent at 280 nm (green, black, blue) and conductivity (red curve) were simultaneously recorded. “
    - Table 1 caption:
    1The enzymes were expressed and purified as described in the Materials and Methods section and the catalytic activity was determined using linoleic and arachidonic acids as a substrate. For each enzyme preparation the amounts of protein added to the assay mixture was normalized to a 1:1 stoichiometry of apoenzyme and non-heme iron.”

Minor points

Figure 3 caption: “(PDB 2P0M entry)” should be “(PDB entry 2P0M)”

Consistence expression for the mutant name in Table 1 (His585Glu and Trp181Glu), 2 (W181E) and 3 (W181E).

Author Response

Reviewer 1.

Reviewer’s comment: I agree that these functional studies support that ALOX15 will act as an allosteric regulator with dimeric formation. Although this study showed functional differences for certain amino acids in α2 and α18 helices, I think that their results can be inferred from previously reported crystal structure and biochemical experiments. Therefore, although I think this manuscript is well written and believe scientifically for the results of the studies conducted by the authors, I believe that this study is at the level of supporting previous studies rather than providing new meaningful information. Meanwhile, I propose the following contents to improve the paper.

Response of authors: Judging the degree of novelty is always a very personal matter. The reviewer is correct when stating that structural heterogeneity of ALOX-isoforms has been suggested before and that these previous reports limit the degree of novelty of the current study. On the other hand, the data presented in our ms are novel and have not been published before. In other words, the basic idea of the paper is not new but the data presented are novel and they provide additional experimental evidence for the validity of the hypothesis that mammalian ALOX-isoforms, in particular ALOX15 orthologs, exhibit a high degree of structural flexibility. The question what exactly is new in this ms can be answered as follows: i) For the first time, we analyzed ALOX-isoforms by native PAGE and the results obtained indicate the existence of different conformers. ii) For the first time, we functionally characterized ALOX15 mutant with possible impact on structural flexibility and the results of these mutagenesis studies characterized the role of certain amino acids for the degree of motional flexibility of the enzyme. iii) For the first time, we carried out MD simulations to explore the structural basis for the mutagenesis induced changes in catalytic properties of ALOX15 and these simulations involved pre-catalytic structures of the enzyme-substrate complexes.

Reviewer’s comment: Under salt-free conditions, Rabbit ALOX15 and human ALOX15B preferred dimers in native gel. On the other hand, it is judged that a weak dimer size was observed even in the condition of salt in the native gel. In protein purification, a buffer without salt was used in gel filtration. Was a dimer observed in this case? I believe that adding the gel filtration results under the same experimental conditions of the native gel can also increase the reliability of the results.

Response of authors: In native PAGE (Fig. 1, Fig. 2A, 2B, 2F) we observed that the protein, which is migrating in SDS-PAGE as single band (Fig. 1A-III, Fig. 2I), travels as double-banded protein. These data indicate that the protein is present in different conformers. These two bands do not necessarily represent monomers and dimers and we never claimed this anywhere in the ms. The native PAGE data simply indicate that the enzymes are present in different conformers, which can be separated from each other on the basis of their different electrophoretic mobilities under native conditions.

In a previous study we have shown that pure rabbit ALOX15 is present in aqueous solutions as a mixture of monomers and dimers and that the monomer/dimer ratio depends on protein concentration, salt concentration and on pH. In gel filtration ALOX15 monomers and dimers are not well separated and it is impossible to separate monomers from dimers if they are present in a mixture of monomers and dimers. In other words, gel filtration is not a suitable method to resolve reversible monomer-dimer mixtures of ALOX15.

Reviewers’ comments: In Figure or Table caption, some contents must be moved to the Method section or deleted.

- Figure 1 caption: ”Rabbit ALOX15 and human ALOX15B were expressed in coli, purified by consecutive affinity chromatography on Ni-NTA resign, anion exchange and size exclusion chromatography and analyzed by PAGE under native and denaturing conditions.”

-“ 200-400 ng protein were applied to precast PAGE gradient gels (4-15%, Bio-Rad, Feldkirchen, Germany). The gel was run for 180 min.”

 - Figure 2 caption: “300 ng pure ALOX protein were applied to precast native PAGE gels, runtime 180 min.”

-“The samples were freshly prepared and maintained in 20 mM Tris-HCl buffer, pH 8.0 (see Methods and Materials).”

- “300 ng pure ALOX15 protein were applied to precast native PAGE gels, runtime was 180 min.” “Absorbance of the column effluent at 280 nm (green, black, blue) and conductivity (red curve) were simultaneously recorded. “

- Table 1 caption: ” 1The enzymes were expressed and purified as described in the Materials and Methods section and the catalytic activity was determined using linoleic and arachidonic acids as a substrate. For each enzyme preparation the amounts of protein added to the assay mixture was normalized to a 1:1 stoichiometry of apoenzyme and non-heme iron.”

Response of authors: As suggested by the reviewer we moved these methodological statements to the methods section.

Reviewer’s comment: Figure 3 caption: “(PDB 2P0M entry)” should be “(PDB entry 2P0M)”

Response of authors: The ms was modified as suggested by the reviewer.

Reviewers’ comments: Consistence expression for the mutant name in Table 1 (His585Glu and Trp181Glu), 2 (W181E) and 3 (W181E).

Response of authors: The names of the mutants in Table 1 have been modified.

Reviewer 2 Report

Conformational Heterogeneity and Cooperative Effects of Mammalian ALOX15

The premise of helix α2 of ALOX15 being conformational dynamic and responsible for enzymatic function is of high importance. The majority of the hypotheses in this paper are based on the reanalysis of the protein structure of rabbit 15-LOX-1 (2POM.pdb). This crystal structure has two molecules in the asymmetric unit and is classified as a heterodimer by the authors. The dimer interface is along helix α2 and α18 between the two protomers. The authors used PISA to analyze helix α18 in the results of the manuscript but did not utilize the same service to analyze the heterodimer interface of rabbit 15-LOX-1. When this heterodimer interface is analyzed by PISA by me it states, “the achieved CSS implies that the interface plays an auxiliary role in complex formation.” This type of caveat should be mentioned in the introduction. Dimers of LOXs appear to be transient in nature and dependent upon protein concentration, temperature, pH, ionic strength, and presence or absence of active site ligands according to the authors. Addressing transient dimer formation of a conditional membrane-binding protein is extremely difficult, and I applaud the scientists for pursuing this goal. I would recommend the authors better explaining in the introduction literature on LOX dimers including “Dimerization of human 5-lipoxygenase”. I would also include “Structural Dynamics of 15-Lipoxygenase-2 via HDX” to address conformational heterogeneity. In the introduction, the authors mention trying to dock AA and LA in protomer A of 2P0M and it requires a rearrangement for binding (unpublished). Why not publish this here? I have a major issue with using protomer B from 2P0M for docking due to the 12-angstrom displacement of helix α2 caused by the ligand RS7. This displacement has not been observed in other LOX structures. I would hypothesize that the bulky inhibitor causes a non-native conformation of rabbit 15-LOX-1 that is stabilized by the other protomer. To quote a paper not referenced in this manuscript by Newcomer and Brash in 2015, “ While it is clear that due to the constriction a conformational change must occur in the enzyme for the substrate to fully enter the active site, whether arachidonic acid binding induces the RS7‐induced conformational change in 15‐LOX‐1 is not clear. The highly homologous 12‐LOX does not require displacement of α2 for its inhibitor, which adopts a U‐like structure, to occupy the site. It is possible that the RS7 conformation is unique to the inhibitor, which is constrained in a conformation inconsistent with the U‐shaped cavity.”

Results section:

While the native PAGE did detect conformational heterogeneity of both LOX isoforms, experimental conditions in the absence of salt are a little disconcerting. The authors should also include the protein concentrations in molarity, so the reader can more easily compare to their past literature [ref 13-15]. I would highly recommend adding the W181E mutant to these experiments since the MD simulations suggest “the interface becomes more open in the mutant enzyme.” Why do MD simulations if you will not validate the findings with a technique you are already using in this manuscript?

I would include more data points in Figure 2.H in the area of greatest change. The authors conclude a steeper rate of change without many data points through that area of the sigmoid to support that statement.

In Figure 3.C. Why does the LA for the H585E mutant stop at 25 uM? Substrate inhibition? Otherwise, the kinetics are beautiful.

Why not include Km in Table 1? Why so many significant figures in Table 1?

For the AA and LA binding studies, why did the authors not use the co-structure of AA to coral 8R-LOX as a benchmark for the in silico studies? Are the authors trying to avoid the “U-shaped” channel model? I would also like to reiterate that using conformer B is an odd choice due to its placement of helix α2. Remember that the highly homologous 12-LOX adopts a conformation more closely resembling 8R-LOX with AA.

Figure 4C color scale is off in the pdf version. Figures 4C and 4D need to be better illustrated for clarity.

Major issues with the manuscript:

The authors mention reference [24] as a previous study, but the reference is a submitted manuscript to another journal. I would mention reference [24] as another study to be peer-reviewed. Separating these two closely MD trajectories and submitting them to separate journals raises a red flag on my end. I hope that the misattributed ref [24] and is supposed to be ref [16].

My understanding that an activated iron ready for hydrogen abstraction from a pentadiene is in the Fe3+-OH state, instead of Fe2+-OH like stated in the paper. If the MD trajectory for the Trp181Glu was run in the wrong oxidation state, the simulations would need to be rerun.

In the discussion, introducing this statement “fluorescence spectra (tertiary structure) were compared (data not shown).” As a reviewer, I need to see the data to make my own conclusions. Usually, unpublished data is not added to the discussion to prove one’s hypothesis.

The “U-shaped” channel model is not mentioned once in this paper. While I understand that the authors might not agree with the “U-shaped” channel model, the competing major hypothesis should at least be acknowledged when talking about docking substrate into the active site.

The activity assays were performed in PBS making any comparisons to the Native PAGE electrophoresis difficult due to the differing amounts of salt. For a more cohesive and convincing story, having the different subsections of the results cross-validate findings is a key to reproducible science.

Author Response

Reviewer 2

Reviewer’s comment: The authors used PISA to analyze helix α18 in the results of the manuscript but did not utilize the same service to analyze the heterodimer interface of rabbit 15-LOX-1. When this heterodimer interface is analyzed by PISA by me it states, “the achieved CSS implies that the interface plays an auxiliary role in complex formation.” This type of caveat should be mentioned in the introduction.

Response of authors: As recommended by the reviewer we added this sentence to the results section.

Reviewers comments I would recommend the authors better explaining in the introduction literature on LOX dimers including “Dimerization of human 5-lipoxygenase”. I would also include “Structural Dynamics of 15-Lipoxygenase-2 via HDX” to address conformational heterogeneity.

Response of authors: We follow the advice of the reviewer and explained the literature describing dimerization of human ALOX5 and the structural dynamics of human ALOX15B in more detail. We also included corresponding references (ref. 16 and 21). It should be stressed at this point, that for human ALOX5 covalent dimerization via the formation of disulfide bridges has been suggested (ref. 16). In contrast, for ALOX15 non-covalent dimers are formed (ref. 14). Thus, for different ALOX paralogs different dimerization mechanisms may exist.

Reviewers’ comments: In the introduction, the authors mention trying to dock AA and LA in protomer A of 2P0M and it requires a rearrangement for binding (unpublished). Why not publish this here?

Response of authors: The docking calculations of AA and LA into monomer A of the crystallographic structure (2P0M) constitute the starting point for our MD simulations. The docking procedure is straightforward but does not represent an independent scientific achievement. In order to publish these findings, the data need to be supported by experimental mutagenesis results. Since such experiments have not been carried out, we do not want to publish the docking studies by their own.

Reviewer’s comment: I have a major issue with using protomer B from 2P0M for docking due to the 12-angstrom displacement of helix α2 caused by the ligand RS7. This displacement has not been observed in other LOX structures. I would hypothesize that the bulky inhibitor causes a non-native conformation of rabbit 15-LOX-1 that is stabilized by the other protomer.

Author’s response: This structural suggestion is plausible but the “non-native character” of the inhibitor bound conformer did neither impact our docking studies nor the MD simulation. In fact, AA and LA were docked into monomer B, which does not involve the inhibitor. To clarify this point, the sentences describing the docking procedure (page 14, line 573). For AA and LA the docking calculations were performed fitting the substrate fatty acids into the substrate binding pocket of the liganded conformer B of rabbit ALOX15. For this purpose, the active site ligand that occupies the substrate-binding pocket of conformer B were removed from the ms and the following statement was introduced: “For AA and LA docking studies the fatty acid substrates were fitted into the substrate binding pocket of the relaxed structure of conformer B of the Trp181Glu-ALOX15 dimer.”

On the other hand, it is not entirely clear whether conformer A really represents a “non-native conformation”. It might well be that helix 2 as a whole is very flexible and when swings away it gives space to allows RS7 to enter the active site. In this scenario, the motional flexibility of helix 2 is the basis for RS7 binding. To stress this point, we included the following paragraph in the Introduction (page 2): „Binding of RS7 at the active site of rabbit ALOX15 is only possible when helix 2 as a whole is displaced (conformer B). Whether this displacement is forced by RS7 binding or whether the intrinsic motional flexibility of helix 2 allows RS7 binding cannot be decided on the basis of the currently available data. It might well be that the intrinsic motional flexibility of helix 2 allows binding of bulky active site ligands. When we overlaid the crystal structures of rabbit ALOX15 (conformer A) with those of liganded human ALOX15B (Figure 1A) and with catalytic domain of the AA 12-lipoxygenating porcine ALOX15 (Figure 1B) we also observed a similar relative position of helix 2. In previous studies we reported that tight association of the PLAT domain and the catalytic subunit of rabbit ALOX15 may be important for stability and integrity of helix 2 [20]. Hydrogen−deuterium exchange mass spectrometry on human ALOX15B also suggested that substrate binding might induce alterations in the structural dynamics of the PLAT domain and of helix 2 [21]. Thus, the conformation of helix 2 observed in conformer B of rabbit ALOX15 might not be considered as “non-native structure”, but be a consequence of the high degree of structural flexibility of the catalytic domain (page 2).

Reviewer’s comments: To quote a paper not referenced in this manuscript by Newcomer and Brash in 2015, “ While it is clear that due to the constriction a conformational change must occur in the enzyme for the substrate to fully enter the active site, whether arachidonic acid binding induces the RS7induced conformational change in 15LOX1 is not clear. The highly homologous 12LOX does not require displacement of α2 for its inhibitor, which adopts a Ulike structure, to occupy the site. It is possible that the RS7 conformation is unique to the inhibitor, which is constrained in a conformation inconsistent with the Ushaped cavity”

Response of authors: We follow the advice of the reviewer and included the corresponding reference into the ms (page 8). However, as discussed above we still disagree with the statement that conformer B represents a” non-native structure”.

Reviewer’s comment: While the native PAGE did detect conformational heterogeneity of both LOX isoforms, experimental conditions in the absence of salt are a little disconcerting. The authors should also include the protein concentrations in molarity, so the reader can more easily compare to their past. I would highly recommend adding the W181E mutant to these experiments since the MD simulations suggest “the interface becomes more open in the mutant enzyme.” Why do MD simulations if you will not validate the findings with a technique you are already using in this manuscript?literature [ref 13-15].

Response of authors: We follow the advice of the reviewer and recalculated the protein concentrations in µM. We also added CD and fluorescence spectra to figure 2 and interpreted the results of these experiments.

Reviewer’s comment: I would include more data points in Figure 2.H in the area of greatest change. The authors conclude a steeper rate of change without many data points through that area of the sigmoid to support that statement.

Author’s response: We repeated this experiment to obtain additional data in the steep part of the denaturation curve. Former Figure 2H is now Figure 2K.

Reviewer’s comment: In Figure 3.C. Why does the LA for the H585E mutant stop at 25 uM? Substrate inhibition? Otherwise, the kinetics are beautiful.

Author’s response: Indeed, we observed a subtle substrate inhibition at substrate concentrations above 25 µM and we briefly mention this finding in the ms.

Reviewer’s comment: Why not include Km in Table 1? Why so many significant figures in Table 1?

Response of authors: In Table 1 we give for each enzyme species numeric values for kcat and kcat/Km. From these two values one can easily calculate Km. However, in response to the suggestion of the reviewer we give the Km values in Table 1 in the revised version of the ms.

Reviewer’s comments: For the AA and LA binding studies, why did the authors not use the co-structure of AA to coral 8R-LOX as a benchmark for the in-silico studies? Are the authors trying to avoid the “U-shaped” channel model? I would also like to reiterate that using conformer B is an odd choice due to its placement of helix α2. Remember that the highly homologous 12-LOX adopts a conformation more closely resembling 8R-LOX with AA.

Response of authors: This criticism is difficult to understand since the reviewer mixes up different ALOX nomenclatures. For human ALOX12 (12-LOX, platelet-type ALOX) there is no complete X-ray structure und thus, this enzyme cannot be used as template for structural modelling and MD simulations. Moreover, the degree of amino acid conservation between human ALOX12 on one hand, and human/rabbit ALOX15 on the other, is rather low and thus, high quality modelling cannot be performed. For coral 8R-LOX a high-quality data set is available, but the degree of amino acid conservation between this enzyme and human/rabbit ALOX15 is also low. Thus, coral 8R-LOX may neither be a good starting point for in silico modelling studies. There is a high-quality X-ray data set for porcine ALOX15, which shares a high degree of amino acid conservation (85%) with rabbit/human ALOX15 and we assume that the reviewer is referring to this enzyme when speaking about 12-LOX. Unfortunately, X-ray data are only available for the catalytic domain. Since our modeling data suggest that substrate binding alters the interdomain interaction of rabbit ALOX15 and since this interdomain interaction may also impact protein dimerization, it was not very helpful to use the X-ray data of porcine ALOX15 (which is by the way an arachidonic acid 12-lipoxygenating enzyme) as starting point for our simulations. Thus, the only choice we had to approach our particular problem was to employ the lower quality X-ray data of complete rabbit ALOX15. Because of the structural problems with monomer A we used monomer B for our modelling approaches. We first carried out a MD trajectory for the Trp181Glu-ALOX15 dimer mutant before docking AA and LA into the cavity of monomer B. Next, long MD simulations of the Trp181Glu-ALOX15/AA and Trp181Glu-ALOX15/LA dimer complexes were run to equilibrate the systems at a given temperature and to explore the binding modes of the two fatty acids inside the binding cavity. After those MD trajectories, any structural artifacts induced by RS7 should have been eliminated. Our MD simulations did not indicate a U-shaped binding mode for AA in the Trp181Glu-ALOX15 dimer mutant. However, the U-shaped model was reproduced when we applied the same methodology on AA binding at the active site of pig ALOX15 and coral 8R-LOX (see references Chemistry. A European Journal 2018, 24, 962-973, ACS Catal. 2017, 7, 4854–4866, respectively).

Reviewer’s comments: But Figure 4C color scale is off in the pdf version. Figures 4C and 4D need to be better illustrated for clarity.

Response of authors: In the submitted pdf version the colors in Fig. 4 C+D are clearly visible. It is most unfortunate that the reviewer cannot see colors, but we do not know the reasons for this. Anyway, for the revised ms we prepared an amended version of Fig. 4 and hope that everything is visible now.

Reviewer’s comment: The authors mention reference [24] as a previous study, but the reference is a submitted manuscript to another journal. I would mention reference [24] as another study to be peer-reviewed. Separating these two closely MD trajectories and submitting them to separate journals raises a red flag on my end. I hope that the misattributed ref [24] and is supposed to be ref [16].

Response of authors: This criticism is fully justified and we apologize for this problem. Reference 16 (in the revised version ref. 17) has been included when referring to the previous MD results of the WT ALOX15/AA dimer complex. Unpublished results (former reference 24) we submit as supplemental file for review process only. Thus, interested readers can inform themselves about the meaning of these unpublished data. Unpublished data does not any more appear in the reference list

Reviewer’s comment: My understanding that an activated iron ready for hydrogen abstraction from a pentadiene is in the Fe3+-OH state, instead of Fe2+-OH like stated in the paper. If the MD trajectory for the Trp181Glu was run in the wrong oxidation state, the simulations would need to be rerun.

Response of authors: The referee is right. This was a typo and it has been corrected. The activated cofactor is now referenced as Fe3+-OH.

Reviewer’s comment: In the discussion, introducing this statement “fluorescence spectra (tertiary structure) were compared (data not shown).” As a reviewer, I need to see the data to make my own conclusions. Usually, unpublished data is not added to the discussion to prove one’s hypothesis.

Response of authors: In response to this criticism, we included the corresponding experimental data to the Figure 2K.

Reviewer’s comment: The “U-shaped” channel model is not mentioned once in this paper. While I understand that the authors might not agree with the “U-shaped” channel model, the competing major hypothesis should at least be acknowledged when talking about docking substrate into the active site.

Author’s response: This criticism is justified and we apologize for being one-sided in this aspect. However, on the basis of this model it is difficult to explain the Triad Concept of ALOX15 specificity. In the revised ms (the results section, page 8) we mention the U-shaped channel model, it states that “Our MD simulations do not indicate a U-shaped binding mode [28] for AA in the Trp181Glu-ALOX15 dimer mutant. In contrast, the U-shaped model was reproduced in previous MD simulations of the pig ALOX15-AA complex [29] and the coral 8R-LOX-AA [30] complex”.

Reviewer’s comment: The activity assays were performed in PBS making any comparisons to the Native PAGE electrophoresis difficult due to the differing amounts of salt. For a more cohesive and convincing story, having the different subsections of the results cross-validate findings is a key to reproducible science.

Response of authors: This is a valid criticism since the conditions in the different experimental setups are not strictly comparable. We know that alterations in the pH change the reaction specificity of arachidonic acid oxygenation by human and rabbit ALOX15 but that salt and temperature alterations hardly impact this enzyme property. Unfortunately, we have not checked whether the reaction kinetics are impacted by salt and pH and this is certainly a limitation of our study. To address this criticism, we included a new paragraph in the results section (page7) in which we discussed this limitation

Round 2

Reviewer 1 Report

The author's response solved some of my concerns. I suggest adding below author's responses for readers and relevant field researchers.

For native gel
"These data indicate that the protein is present in different conformers. These two bands do not necessarily represent monomers and dimers. The native PAGE data simply indicate that the enzymes are present in different conformers, which can be separated from each other on the basis of their different electrophoretic mobilities under native conditions.
In a previous study we have shown that pure rabbit ALOX15 is present in aqueous solutions as a mixture of monomers and dimers and that the monomer/dimer ratio depends on protein concentration, salt concentration and on pH. In gel filtration ALOX15 monomers and dimers are not well separated and it is impossible to separate monomers from dimers if they are present in a mixture of monomers and dimers. In other words, gel filtration is not a suitable method to resolve reversible monomer-dimer mixtures of ALOX15."

*In addition, even if the monomer-dimer is not separated in size exclusion chromatography (SEC), I request to add the SEC results using the buffer under the same conditions as native gel loading. When these data are added, it can be agreed that the approach with native gel is new as mentioned by the authors.

Minor
-The author used various expressions such as'helix α2','helix 2', and'α2 helices'. These expressions should be unified as one.

-line 128: For'Mini-PROTEAN® TGXTM precast gels' it would be appropriate to transfer to the method section.

-line 166-167: "His585 (conformer A, helix α18) and Trp181 (conformer B, 167 helix α2) contribute to the inter-monomer contact plane in the crystal structure of rabbit 168 ALOX15 with their buried surface area of ​​86.65 and 153.82 Å, respectively...with their buried surface area of ​​86.65 and 153.82 Å, respectively." Is the buried surface for residue or helix? On the other hand, because it is an area, the unit must be modified to Å2.

Author Response

Reviewer 1

Comment of reviewer: The author's response solved some of my concerns. I suggest adding below author's responses for readers and relevant field researchers.

For native gel

"These data indicate that the protein is present in different conformers. These two bands do not necessarily represent monomers and dimers. The native PAGE data simply indicate that the enzymes are present in different conformers, which can be separated from each other on the basis of their different electrophoretic mobilities under native conditions. In a previous study we have shown that pure rabbit ALOX15 is present in aqueous solutions as a mixture of monomers and dimers and that the monomer/dimer ratio depends on protein concentration, salt concentration and on pH. In gel filtration ALOX15 monomers and dimers are not well separated and it is impossible to separate monomers from dimers if they are present in a mixture of monomers and dimers. In other words, gel filtration is not a suitable method to resolve reversible monomer-dimer mixtures of ALOX15."

Response of authors: We do agree with the reviewer that interpretation of the native PAGE data should be included into the ms and we inserted a corresponding paragraph into the Discussion section (page 12, line 455-470). This paragraph now reads: “The observation that our enzyme preparation, which migrates as a single protein band in SDS-PAGE, is heterogenous (two protein bands) in native PAGE (Fig. 2A+B) suggests that the enzyme is present in two different conformers, which can be separated from each other on the basis of their distinct electrophoretic mobilities under native conditions. In a previous study [15] we have shown that pure rabbit ALOX15 is present in aqueous solutions as a reversible monomer-dimer equilibrium. This equilibrium depends on the ALOX15 concentration, presence and absence of salt and on pH [15]. Dilution of the enzyme solution strongly favor monomer formation. In the present study the enzyme concentration is below 10 µM. Under these experimental conditions monomers strongly prevail and dimers are virtually absent. Thus, the two bands observed in native PAGE do not represent monomers and dimers but rather two different monomeric enzyme conformers. To further support this conclusion, we performed size exclusion chromatography with our enzyme preparations (Fig. 2F, left panel). Here we observed identical retention volumes and the estimated hydrodynamic radius of the wt protein is identical with the corresponding values of the mutant enzymes. Here again, no protein dimers were observed, which should be present as front shoulders of the monomer peak.”

Comment of reviewer: In addition, even if the monomer-dimer is not separated in size exclusion chromatography (SEC), I request to add the SEC results using the buffer under the same conditions as native gel loading. When these data are added, it can be agreed that the approach with native gel is new as mentioned by the authors.

Response of authors: Following the advice of the reviewer we added SEC data to Fig. 2 (panel F).

Comment of reviewer: -The author used various expressions such as'helix α2','helix 2', and'α2 helices'. These expressions should be unified as one.

Response of authors: Following the advice of the reviewer, we synchronized our wording.

Comment of reviewer: line 128: For'Mini-PROTEAN® TGXTM precast gels' it would be appropriate to transfer to the method section.

Response of authors: Following the advice of the reviewer, we removed this sentence. This description is present in the methods section.

Comment of reviewers: line 166-167: "His585 (conformer A, helix α18) and Trp181 (conformer B, 167 helix α2) contribute to the inter-monomer contact plane in the crystal structure of rabbit 168 ALOX15 with their buried surface area of ​​86.65 and 153.82 Å, respectively...with their buried surface area of ​​86.65 and 153.82 Å, respectively." Is the buried surface for residue or helix? On the other hand, because it is an area, the unit must be modified to Å2.

Response of authors: The values are given for buried surface areas of the residues. We apologize for this misprint which has been corrected in the revised version of the ms.

Reviewer 2 Report

The authors vastly improved their manuscript from the comments and suggestions. While I might not fully agree with the conclusions, I believe that competing hypotheses with data to support their claims should be published. 

I am still confused by figure 4. (B) I see a light blue and purple color on my screen instead of a turquoise. (C and D) I understand now that the colors represent the fatty acids. The protein structures have been superimposed (not clearly stated)? I see secondary structure elements that are light green and another that appears light grey or white for the other structure (not purple or light purple/lavender).

Author Response

Reviewer 2

Comment of reviewer: I am still confused by figure 4. (B) I see a light blue and purple color on my screen instead of a turquoise. (C and D) I understand now that the colors represent the fatty acids. The protein structures have been superimposed (not clearly stated)? I see secondary structure elements that are light green and another that appears light grey or white for the other structure (not purple or light purple/lavender).

Response of authors: We apologize for the misleading indications in the Figure 4 caption. They were corrected in the revised version of the ms.

Round 3

Reviewer 1 Report

The author's response solved my concerns. I recommed to accept this manuscript after revising the page number for this manuscript's reference.